# Exploring the Impact of a Supportive Work Environment on Chinese L2 Teachers’ Emotions: A Partial Least Squares-SEM Approach

**DOI:** 10.3390/bs14050370

**Published:** 2024-04-28

**Authors:** Yonghong Zeng, Jiaying Yu, Hanwei Wu, Wentao Liu

**Affiliations:** School of Foreign Studies, Hunan Normal University, Changsha 410000, China; yhzeng@hunnu.edu.cn (Y.Z.); jyyucs@hunnu.edu.cn (J.Y.); 2021100172@jisu.edu.cn (W.L.)

**Keywords:** L2 teachers’ emotions, perceived climate, supervisory relationship, peer group interaction, perceived organization support

## Abstract

Second language (L2) teachers’ emotions can influence their well-being and students’ performance. However, most of the existing studies have focused on the role of individual factors in affecting L2 teachers’ emotions, while leaving environmental factors underexplored. To fill this gap, this study aimed to examine how the four dimensions of a supportive work environment (SWE) (perceived climate, PC; supervisory relationship, SR; peer group interaction, PGI; and perceived organization support, POS) relate to L2 teachers’ emotions (enjoyment, anxiety, pride, and anger). A sample of 406 Chinese L2 teachers completed two valid scales to measure their SWE and emotions. The data were analyzed by Partial Least Squares-Structural Equation Modeling (SEM) using Smart PLS 3 software. The results showed that (1) PC, PGI, and POS had a positive and significant effect on enjoyment, while SR had no significant effect; (2) PGI and POS had a negative and significant effect on anxiety, while PC and SR had no significant effect; (3) PGI had a positive and significant effect on pride, while the other three dimensions had no significant effect; and (4) POS had a negative and significant effect on anger, while the other three dimensions had no significant effect. The study concludes with some implications for L2 teachers’ education.

## 1. Introduction

Emotions are ubiquitous in second language acquisition (SLA) and have a significant impact on L2 performance. Previous research on L2 emotions has mainly focused on the negative aspects, especially anxiety [1]. However, in recent years, the field of SLA has witnessed a shift of focus from negative emotions to positive emotions, thanks to the development of positive psychology and its application to SLA [2]. This shift has enabled language researchers to explore L2 emotions from a more holistic perspective and to examine the role of positive emotions in SLA processes and achievements [3,4,5].

However, compared with learners’ emotions, teachers’ emotions have been largely neglected by language researchers, despite their importance in educational contexts. Previous studies in general education have shown that teachers’ emotions have various implications for their own well-being [6,7], burnout [8], and work engagement [9,10], as well as for students’ learning outcomes [11]. This is because teachers’ emotions can influence the quality of their teaching and interaction with students [12,13]. Therefore, it is essential to explore the emotional experiences of L2 teachers and the factors that shape them.

According to Dewaele, et al. [14], such explorations have emerged since 2016. They have mainly examined the antecedents and consequences of L2 teachers’ emotions [7,15,16]. However, the majority of the studies have focused on the internal factors of L2 teachers, such as self-efficacy, resilience, L2 grit, etc. [17,18], while neglecting the external factors. The control–value theory (CVT) suggests that emotions are influenced by control–value appraisals like proximal antecedents, and by other distal antecedents, including individual and environmental ones [19].

Considering that the environmental antecedents of L2 teachers’ emotions have received little attention in previous studies, especially in the Chinese context, this study, based on the CVT, aimed to examine the role of a supportive work environment (SWE), an archetypical environmental factor, in shaping L2 teachers’ emotions in China. 

## 2. Literature Review

### 2.1. L2 Teachers’ Emotions

Among research on L2 emotions, the CVT, addressing the antecedents and outcomes of achievement emotions, is one of the most frequently employed theories [19]. According to the CVT, achievement-related emotions are defined as emotional states directly linked to achievement-oriented actions, such as teaching, or the outcomes thereof, specifically success and failure. In the CVT, eight primary emotions linked to student achievement are highlighted: enjoyment, hope, pride, anxiety, anger, hopelessness, shame, and boredom. These emotions are organized into a three-dimensional model based on valence, activation, and object focus. Valence distinguishes between positive and negative emotions; for example, the positive feeling of enjoyment versus the negative feeling of anxiety. Activation separates emotions that are physiologically arousing from those that are calming, such as the stimulating emotion of hope compared to the tranquilizing effect of hopelessness. Object focus categorizes emotions based on their relation to activities or outcomes, like the activity-associated emotion of boredom versus the outcome-associated emotion of shame.

Among teachers, the emotions of enjoyment, pride, anxiety, and anger are reported to be commonly experienced [20,21]. Besides, the CVT posits that individuals’ emotional experiences are shaped by their perceived control over, and the value they attribute to, tasks and outcomes they deem subjectively meaningful [19]. This theory suggests that the proximal antecedents to emotions are control–value appraisals. For example, when individuals discern a substantial level of control and ascribe high value to a pursuit, they are predisposed to encounter positive emotions such as enjoyment. Conversely, the experience of reduced control and a low appraisal of value may evoke negative emotions like hopelessness. Despite the decisive role of control–value appraisals, other distal antecedents are also acknowledged by the CVT, which could be roughly divided into individual antecedents and environmental antecedents. In this study, we focused on the latter.

#### 2.1.1. Enjoyment

Enjoyment is a positive, activating, and activity-linked emotion that emerges from the feeling of pleasure and satisfaction that learners experience when they participate in or complete a learning task. Enjoyment motivates individuals to overcome challenges and improve their skills [22], and it is vital for language teaching and learning [23]. However, the influence of enjoyment on the learning process remains contentious. According to the resource allocation model proposed by Ellis and Ashbrook [24], positive emotions like enjoyment could result in a surge of intrusive thoughts. Such thoughts could occupy cognitive resources [25], which may reduce the amount of effort dedicated to ongoing cognitive tasks [26]. This reduction in effort could potentially hinder the learning process. Enjoyment for teachers could result from the learners’ outstanding performance in the class, as Frenzel [27] stated that teachers’ expectations for learners’ practice are one of the teachers’ emotional antecedents. Furthermore, teachers’ enjoyment reduces their negative emotions and protects them from emotional exhaustion [11].

#### 2.1.2. Anxiety

Anxiety is a negative, activating, and outcome-associated emotion that involves feelings of fear, nervousness, or worry, especially about the future [28]. Language anxiety, in particular, is influenced by various factors, such as cognitive and emotional processes, situational demands, and social interactions, which can vary over time [29]. For nearly forty years, since Horwitz [1]’s foundational study, the prevalence of anxiety among L2 learners has been a subject of research. While a handful of studies suggest that anxiety might enhance learning in some learners by fostering an extrinsic motivation to exert effort and avoid failure [30], anxiety overall tends to negatively impact the L2 performance of most students, as outlined in a previous meta-analysis [31]. According to Frenzel [27], teachers tend to experience less anxiety than students because they do not frequently encounter challenges or failures in their language learning. Nonetheless, novice teachers may face higher levels of anxiety due to the complexity and uncertainty of learning how to teach and establishing relationships with students and guardians [32].

#### 2.1.3. Pride

Pride is a positive, activating, and outcome-related emotion that stems from individuals’ positive self-evaluation of their own attributes, achievements, or affiliations [33]. Pride can be elicited by attaining a goal, surmounting a challenge, or exhibiting competence or excellence in a domain [34]. In the educational context, pride is a key emotion in the classroom and is commonly experienced by both teachers and students [35]. Previous research has confirmed the benefits of pride to students’ L2 performance [4]. However, findings in educational psychology suggest that, in some situations, positive emotions such as pride might lead to overconfident evaluations [19]. This can result in shallow processing of information and a decreased drive to engage with difficult objectives, potentially lowering the likelihood of achieving success. For teachers, pride has been found to have benefits in terms of self-esteem, social status, and well-being [36,37]. Within language-teaching contexts, language teachers’ pride plays a significant role in enhancing students’ positive language learning experiences [38].

#### 2.1.4. Anger

Anger is a negative, activating, and activity-connected emotion that can lead to aggression and is influenced by the perception of being wronged and the level of arousal [39]. Suls [40] notes that anger is both an aspect of aggression and a personality characteristic linked to sensations of mistreatment and arousal levels. While research has shown a negative correlation between student anger and academic achievement [41], it has also been observed that anger can drive some students to excel in L2 learning as a form of retaliation against perceived injustices by school authorities [42]. Teachers’ anger may arise from discrepancies between their expectations and students’ achievements or behavior, or due to interactions with difficult students, parents, colleagues, or the educational system [43]. Similarly, teacher anger can have dual outcomes; while frequent anger may hinder student participation, authentic displays of anger can lead to increased engagement [44].

### 2.2. Supportive Work Environment

Within the framework of the CVT, environmental antecedents for students’ emotions such as autonomy support [45], teachers’ enthusiasm [46], peer feedback [47], etc. have been well recognized. In contrast, such explorations have been surprisingly scant concerning teachers, although teachers’ emotions have profound effects on both their own well-being and their teaching quality [48,49]. Teachers are employees of educational institutions so it is advisable to put this population into the field of organizational psychology. A supportive work environment (SWE) serves as a precursor for employees’ emotions, encompassing the workplace climate characterized by support from supervisors and peers, the presence of challenges, and the opportunity to implement skills and knowledge acquired [50]. Previous literature on organizational psychology has shown that SWEs could be measured by the following four aspects: perceived climate (PC) [51], supervisory relationship (SR) [52], peer group interaction (PGI) [53], and perceived organizational support (POS) [54].

#### 2.2.1. Perceived Climate

PC denotes the collective beliefs and significance employees ascribe to their organization’s policies, practices, and procedures they encounter, along with the behaviors they notice being recognized, encouraged, and anticipated in relation to the organization’s human resources [55,56,57]. Research indicates that a favorable PC is linked to reduced teacher attrition [58], job satisfaction [59], and improved emotional well-being [60]. Conversely, a negative PC can lead to adverse outcomes. Additionally, there is a suggestion from prior research that PC may be intricately connected to teachers’ emotions [60]. Yet, detailed explorations into the effect of PC on multiple teacher emotions are limited. This aspect might be particularly critical for L2 teachers who frequently interact with students from varied cultural backgrounds. A nurturing educational environment is pivotal in fostering a respectful and inclusive atmosphere that appreciates cultural diversity, which is vital in the context of language education.

#### 2.2.2. Supervisory Relationship

SR pertains to the emotional connection and mutual understanding between a supervisor and an employee regarding the tasks and objectives of supervision [61]. Within educational institutions, the SR between school administrators and teachers is crucial for the teachers’ performance and their continuous professional development. This dynamic facilitates a cooperative atmosphere conducive to goal-setting and enriches teachers’ understanding of student learning dynamics. As a result, it acts as a catalyst for improvements in their professional roles [62,63]. Although prior research has acknowledged the significance of emotions in the workplace [64], studies providing insights into the work environment factors that influence employees’ diverse emotions are scarce. This is particularly true for studies in educational contexts that concentrate on SR, one component of work environment.

#### 2.2.3. Peer Group Interaction

PGI involves the communication and influence processes among a group of peers who have some common attributes, such as age, social status, economic status, occupation, or education [53]. In the realm of education, prior studies have indicated that PGI among educators fosters mutual professional growth. This is achieved through promoting introspection on pedagogical methods, forming a community of professional dialogue, elevating teaching quality standards, and encouraging cooperative efforts [65]. While PGI proves advantageous across various fields, it is particularly crucial for L2 instructors due to the inherent demands of language acquisition, which include ongoing practice, feedback, and adjustments. However, the extent to which PGI provokes emotional responses in L2 educators remains an area that is not well examined.

#### 2.2.4. Perceived Organization Support

POS is a key concept in organizational research. POS is defined as “employee beliefs concerning the extent to which the organization values their contribution and cares about their well-being” [66]. POS is rooted in social exchange theory, which suggests that employees attribute human-like qualities to their organizations and develop overarching perceptions of how much their organization values their well-being [67]. For teachers, POS is particularly influential, affecting their professional lives profoundly. Research indicates that teachers who perceive strong support from their institutions often experience numerous benefits, such as teachers’ self-efficacy and motivation, and coping with job-related stress [68,69]. However, the literature has yet to investigate the potential impact of POS on teachers’ emotions.

### 2.3. Related Empirical Studies

The role of a SWE in influencing L2 teachers’ emotions has not been comprehensively examined in previous studies, although some of its components have been entailed in some research. These studies mainly adopted qualitative methods to investigate how L2 teachers’ emotions were related to peer group interaction, perceived organizational support, and perceived school climate, which are three dimensions of a SWE. For instance, Cowie [70] interviewed nine experienced L2 teachers with diverse teaching backgrounds and found that their emotions were affected by the quality of their collegial relationships and the availability of institutional support. Burić and Frenzel [71] also employed interviews to examine the sources of anger among 25 teachers, including L2 teachers. They revealed that inadequate organizational support, such as frequent changes in the curriculum, excessive paperwork, and poor material conditions, was the trigger of anger. In a similar vein, Sun and Yang [72] conducted a case study with two senior high school L2 teachers using interviews and reflection logs and revealed that their emotions were influenced by their collaboration and interactions with colleagues and supervisors. A recent quantitative study by Zhang, Fathi, and Mohammaddokht [17] also involved a SWE to some extent by examining the impact of perceived school climate, another dimension of a SWE, on the enjoyment of 355 teachers with different teaching grades. Using Structural Equation Modeling (SEM), they confirmed that perceived school climate had a significant and positive effect on teachers’ enjoyment. 

These studies suggest that a SWE may play an important role in shaping L2 teachers’ emotions, but they do not cover all the aspects of SWEs. Therefore, a more comprehensive and systematic study is needed to investigate the effects of the four dimensions of a SWE on L2 teachers’ emotions. The following research hypotheses were tested:

**H1.** *SWE (PC, SR, PGI, POS) could significantly and positively predict Chinese L2 teachers’ enjoyment*.

**H2.** *SWE (PC, SR, PGI, POS) could significantly and negatively predict Chinese L2 teachers’ anxiety*.

**H3.** *SWE (PC, SR, PGI, POS) could significantly and positively predict Chinese L2 teachers’ pride*.

**H4.** *SWE (PC, SR, PGI, POS) could significantly and negatively predict Chinese L2 teachers’ anger*. 

## 3. Materials and Methods

### 3.1. Participants

This study employed convenience sampling to select 435 college L2 teachers from various provinces of China, such as Hunan, Zhejiang, Henan, and so on, as the initial sample for the quantitative data collection (see Table 1). However, 29 teachers (6.65%) were eliminated from the analysis based on the exclusion criteria: (1) they gave identical responses to all items; (2) they were incomplete. The final sample consisted of 406 participants. The demographic information of the participants is demonstrated in Table 1. Participants received information on the survey’s aim, how to fill out the questionnaire, and the assurance of confidentiality and anonymity before the survey began. They also had the opportunity to raise any queries about the survey and agree to participate. 

### 3.2. Instruments

#### 3.2.1. Supportive Work Environment (SWE)

A SWE had been scaled on four dimensions, including PC [51], SR, PG [53], and POS [54]. POS was measured on eight items. An example item on POS contains “My school cares about my well-being”. PC was measured on three items. One item of the scale on PC includes “English teachers are treated with respect in my school”. The SR was measured on seven items. One of the items for SR constitutes “Supervisor is reliable and trustworthy”. PG was measured on eight items. For measuring PG, one example of an item includes “I socialize with co-workers even outside the job”. Following Naz, Li, Nisar, Khan, Joo, and Anwar [50], all these dimensions were measured on a 5-point Likert scale ranging from 1 (strongly disagree) to 5 (strongly agree). A higher score indicates a better SWE. 

#### 3.2.2. Achievement Emotions Questionnaire-Teachers (AEQ-T)

To assess the four achievement emotions of L2 teachers (anxiety, pride, enjoyment, and anger), we used the AEQ-T developed by Hong, Nie, Heddy, Monobe, Ruan, You, and Kambara [20]. For measuring anxiety, one example of the items includes “I feel uneasy when I think about teaching”. One item of the scale on pride includes “I am proud of the way I am teaching”. One of the items for enjoyment constitutes “I generally enjoy teaching”. An example item on anger contains “Sometimes I get really mad while I teach”. The questionnaire contains 15 items, each rated on a 4-point Likert scale from 1 (strongly disagree) to 4 (strongly agree). Higher scores indicate stronger emotions. 

### 3.3. Data Collection

We contacted several L2 teachers from colleges in China and asked them to send the link of an online questionnaire survey supported by Wenjuanxing (www.wjx.cn, accessed on 20 April 2023).

Through snowball sampling, we obtained quantitative data from 406 L2 teachers. The teachers received informed consent forms that outlined the study’s procedures, ensuring confidentiality and anonymity. To validate the self-report scales related to a SWE and emotions, a translation and back-translation method was utilized. The scales were initially translated from English to Chinese and then retranslated to English by three bilingual researchers. Subsequently, a psychology expert in translation examined and refined the items’ phrasing to achieve the highest possible semantic consistency between the English and Chinese versions. Participants were asked to fill out the scales in Chinese, but the English versions were also made available to them for reference to the original item meanings as needed.

### 3.4. Data Analysis

This study utilized the Structural Equation Modeling (SEM) technique for data analysis, which encompasses two primary branches: covariance-based SEM (CB-SEM) and Partial Least Squares-SEM (PLS-SEM). Historically, CB-SEM has been the prevalent method in scholarly investigations. However, PLS-SEM has gained prominence due to its unique statistical attributes, positioning it as a viable substitute for CB-SEM [73]. The principal aim of this study was to elucidate and predict the constructs under investigation, grounded in theoretical frameworks. In comparison with CB-SEM, which requires a global goodness-of-fit criterion, PLS-SEM is particularly apt for this task, given its emphasis on the explanation and prediction of variables. In recent times, the research community has shown a growing preference for consistent PLS-SEM, drawn by its merits. This advanced variant upholds the core strengths of traditional PLS-SEM while delivering supplementary benefits, including a reduced parameter estimation bias and an enhanced predictive analysis structure [74]. Therefore, we adopted consistent PLS-SEM to assess both the measurement and structural models, employing the Smart PLS 3 software for this purpose.

## 4. Results

### 4.1. Testing Common Method Bias

Common method variance (CMV) refers to the variance that is attributed to the measurement method rather than the construct that the measures are supposed to represent [75]. CMV can be problematic in research when the independent and dependent variables are measured by the same person’s self-reported data [76]. One way to detect CMV is to use the variance inflation factor (VIF) test, which indicates the degree of multicollinearity among the variables. A VIF value higher than 3.3 suggests that the model may be affected by CMV, and the corresponding items should be removed. In this study, the VIF test showed that the items POS1, POS6, SR3, SR5, and SR7 had VIF values higher than 3.3, so they were excluded from the model. 

### 4.2. Testing the Measurement Model

Construct validity refers to the extent to which the measure captures the actual concept as theorized [77]. Ramayah, et al. [78] suggested that any item with a loading higher than 0.5 on two or more components should be considered as having significant cross-loadings. Based on this criterion, the items Anxiety 4, PGI 4, and PGI 7 were removed. Then, the PLS algorithm analysis was conducted again to obtain the new loadings and cross-loadings, as shown in Table 2. The results indicated that all the items of a specific construct had high loadings only on their own construct, and low loadings on the other constructs, thus confirming the construct validity.

Convergent validity assesses whether the items that are related to a specific variable share a high proportion of variance and can be evaluated by average variance extracted (AVE) and composite reliability (CR) [77]. The results showed that the AVE values ranged from 0.589 to 0.813, indicating that each construct explained more than 50% of the variance of its items. CR reflects the extent to which the latent construct is measured reliably by the observed items [79]. In this study, the CR values ranged from 0.835 to 0.929, which were higher than the threshold of 0.7 [80]. Therefore, this study confirmed the convergent validity of the constructs, as shown in Table 2, which presented the findings of the measurement model. Moreover, the CR values demonstrate that the measurement scales were reliable (>0.7) [79]. 

To assess the discriminant validity, we examined the extent to which each latent variable was distinct from the others, following the approach of Hair, Black, Babin, and Anderson [77]. This study adopted the multitrait-multimethod matrix proposed by Henseler et al. and used the heterotrait–monotrait (HTMT) ratio as the criterion to evaluate the discriminant validity [81]. According to Kline, the HTMT ratio should be less than 0.85 for each pair of latent variables to indicate sufficient discriminant validity [82]. Table 3 shows the HTMT ratios for the measured variables in this study. All the values were below the threshold of 0.85, suggesting that there was no issue of discriminant validity in the model.

Moreover, we utilized multiple indices to evaluate the approximate fit of our model, including the standardized root mean square residual (SRMR), normed fit index (NFI), and measures of exact fit (d_ULS and d_G). According to the Smart PLS 3 software, the SRMR value was 0.064, under the 0.08 benchmark [83]. The NFI was recorded at 0.92, above the 0.9 standard [84], while both d_ULS and d_G fell within the 95% confidence interval during bootstrapping [85]. These indicators collectively suggest that the data fits well with our proposed model.

### 4.3. Testing the Structural Model

The study conducted descriptive statistics and correlation analysis on the data of the measurement model before testing the structural model. Table 4 reports the mean values, standard deviations, and correlation coefficients of all the variables in the study. The mean values of all constructs range from 2.505 to 3.888, indicating the average level of responses for each variable. The correlation coefficients show the strength and direction of the linear relationships between the variables, as shown in Table 4. All the variables have significant correlations with each other.

The results show that the four dimensions of SWEs have significant correlations with the four emotions. The correlations are positive for enjoyment and pride and negative for anxiety and anger. Table 4 displays the correlation coefficients for each pair of variables. 

The highest correlation is between PC and enjoyment (r = 0.517, *p* < 0.05), followed by POS and enjoyment (r = 0.493, *p* < 0.05). The lowest correlation is between SR and anger (r = −0.218, *p* < 0.05), followed by PGI and anxiety (r = −0.225, *p* < 0.05). These findings confirmed the suitability of the data for PLS-SEM analysis.

We then tested the structural model by applying the bootstrapping technique to evaluate the goodness-of-fit of the theoretical model by examining the R^2^ values of the dependent variables and the significance of the path estimates [86]. Table 5 presents the R^2^ values and the effect sizes of the four dimensions of SWEs on the four emotions. Thus, hypotheses were addressed. The results show that:

**H1.** *SWE explained 32.9% of the variance in enjoyment. Among the four dimensions, PC (β = 0.233, p < 0.05), PGI (β = 0.247, p < 0.001), and POS (β = 0.173, p < 0.05) had a positive and significant impact on enjoyment, while SR had no significant effect*.

**H2.** *SWE explained 15.6% of the variance in anxiety. Among the four dimensions, only PGI (β = −0.211, p < 0.05) and POS (β = −0.231, p < 0.05) had a negative and significant impact on anxiety, while PC and SR had no significant effect*.

**H3.** *SWE explained 7.4% of the variance in pride. Among the four dimensions, only PGI (β = 0.163, p < 0.05) had a positive and significant impact on pride, while the other three dimensions had no significant effect*.

**H4.** *SWE explained 10.8% of the variance in anger. Among the four dimensions, only POS (β = 163, p < 0.05) had a negative and significant impact on anger, while the other three dimensions had no significant effect*.

## 5. Discussion

Emotions in the workplace significantly influence employees’ well-being [68,69]. While some studies have hinted at the impact of a SWE on employees’ emotions, a comprehensive exploration of its influence on multiple emotional responses remains unexplored [70,71,72]. L2 teachers, in particular, warrant special consideration. Their emotional state is not only pivotal to their personal well-being but also plays a critical role in their students’ L2 learning [49]. As language serves as the fundamental medium of communication, and emotions are deeply embedded in communicative exchanges, L2 teachers must adeptly manage both linguistic challenges and the emotional subtleties inherent in cross-cultural interactions [2]. Therefore, to inform the design of targeted interventions, this study examined the effects of a SWE on L2 teachers’ emotions by examining four dimensions of a SWE: PC, SR, PGI, and POS. These dimensions were expected to influence four types of emotions: enjoyment, anxiety, pride, and anger. Initially, the construct validity and reliability of the instruments used in the study were confirmed. Then, the study applied consistent PLS-SEM to test the structural model and examine the causal relationships among the variables.

In terms of H1, this study demonstrated that a SWE could enhance the enjoyment of L2 teachers, explaining nearly 30% of the variance. The three dimensions were PC, PGI, and POS, which suggests that EFL teachers experience more enjoyment when they work in a friendly and rewarding environment, maintain rapport with their peers, and receive assistance from the organization. This finding corroborates the result of Zhang, Fathi, and Mohammaddokht [17], who identified perceived school climate as a promoter of L2 teachers’ enjoyment. This finding can also be rationalized by the strong link between a SWE and job satisfaction and teacher self-efficacy. Previous research has shown the significant impact of a favorable work environment on teacher burnout [87,88] and teachers’ job satisfaction [89,90]. It can be inferred that a SWE can foster enjoyment by cultivating a sense of satisfaction and emotional exhaustion among L2 teachers. However, their enjoyment was not influenced by their relationship with supervisors. One possible reason is that China has a high-power distance culture, in which supervisors are minimally involved in the regular work of L2 teachers, which hinders developing rapport between them [91,92]. 

Concerning H2, we discovered that a SWE could reduce the anxiety levels of L2 teachers, accounting for approximately 15% of the variation. However, the different dimensions of a SWE had varying degrees of influence. The findings indicated that PGI and POS were more effective than PC and SR in mitigating anxiety. This suggests that L2 teachers appreciate the feedback and recognition from their peers and institutions more than the general climate and relationship with their supervisors. This result aligns with the previous research by Cowie [70] and Sun and Yang [72], which showed that L2 teachers’ emotional well-being could be enhanced by friendship, respect, and collaboration with their colleagues. A plausible explanation for the insignificance of PC in predicting L2 teachers’ anxiety is that the general climate is too ambiguous to have a stable impact. General climate can encompass various factors, such as the curriculum, the policies, the expectations, the norms, and the values of the schools. These factors may differ across contexts, and may not be perceived similarly by different L2 teachers. Regarding the negligible effect of SR, we also attributed it to the possible estrangement caused by the high-power distance between L2 teachers and their supervisors [91,92].

Regarding H3, this study also revealed that a SWE could increase the pride of L2 teachers, accounting for only about 7% of the variance. However, the influence of different dimensions of a SWE varies. The finding indicates that only PGI had a positive and significant impact on pride, while the other three dimensions (PC, SR, and POS) had no significant effect. This suggests that L2 teachers feel prouder of their work when they have interactions with their colleagues. This result partially aligns with the study by Mairitsch, Sulis, Mercer, Mairi, and Shin [36], which showed that L2 teachers’ sense of pride is socially constructed. L2 teachers’ pride elicited by PGI may have two sources. On the one hand, pride is a self-conscious emotion, which can be divided into self-based and social comparison-based aspects [34,93]. L2 teachers may experience pride when they feel they are superior to their peers in terms of their performance, reflecting the social comparison-based facet of pride. On the other hand, the social identity theory suggests that one’s pride is influenced by the identification with a relevant group [94,95]. Therefore, L2 teachers who have high PGI may identify more with their profession and feel prouder of their achievements. PC, SR, and POS seem less relevant to the nature of pride.

As for H4, we confirmed that a SWE could mitigate the anger of L2 teachers, but only accounting for around 10% of the variance. However, only POS has a negative and significant impact on anger, while the other three dimensions (PC, SR, and PGI) have no significant effect. This means that L2 teachers feel less angry at their work when they perceive that their organization values their contribution, cares about their well-being, and provides them with adequate resources and opportunities. This finding is consistent with the previous research by Burić and Frenzel [71], which showed that L2 teachers’ poor POS was a trigger of L2 teachers’ anger. A possible explanation for the insignificance of the other three dimensions of SWE is that they are less relevant to the nature of anger. Anger is often induced by a perceived injustice or mistreatment from a powerful source [96,97]. Therefore, L2 teachers may be more sensitive to the support from their organization, which represents the authority and legitimacy in their work context, than the climate, relationship, and interaction with their colleagues and supervisors.

This study makes several theoretical contributions. Firstly, this study, to the best of our knowledge, is the first study that examined the impact of all dimensions of a SWE on L2 teachers’ emotions. In addition, our study applied the principles of the CVT to the research on L2 teachers’ emotions, thus extending its applicability. CVT is a comprehensive framework that integrates various antecedents and outcomes of learners’ emotions, but it has not been widely used in the context of L2 teaching. Our study provides empirical evidence for the validity and usefulness of CVT in explaining L2 teachers’ emotional experiences.

This study has implications for education. It suggests that L2 teachers can enjoy their work more if they have better PC, PGI, and POS. Therefore, school leaders should create a supportive atmosphere where L2 teachers could feel valued. L2 teachers should also communicate with their peers to increase their enjoyment. Second, this study indicates that L2 teachers can reduce their anxiety if they have PGI and POS. Hence, school administrators should provide adequate resources for L2 teachers to help them overcome the work challenges and ease their anxiety. L2 teachers should also seek support from their colleagues to cope with stress and uncertainty. Third, this study shows that L2 teachers can enhance their pride if they have PGI. Thus, L2 teachers should build good relationships with their colleagues to improve their self-image and pride. They should also recognize their own and others’ achievements to appreciate their work and its value. Fourth, this study reveals that L2 teachers can lower their anger if they have POS. Therefore, supervisors should care for L2 teachers, such as listening to their needs and helping to solve their problems. This can help L2 teachers feel respected and appreciated, thus mitigating their anger. 

This study also has some limitations that should be acknowledged. First, this study only incorporated Chinese samples, so the generalizability of our findings to other cultures is uncertain. A cross-national comparative study could be conducted in the future to examine the cultural differences in L2 teachers’ emotions and their antecedents. Second, the cross-sectional design of this study presents constraints in assessing the longitudinal influence of SWE on L2 teachers’ emotions. The absence of temporal analysis highlights the imperative for future longitudinal investigations to capture the evolving dynamics. Third, this study did not examine the impact of other environmental antecedents, such as students’ feedback, on L2 teachers’ emotions. Hence, future studies could probe the impact of other external factors to present a more comprehensive picture of L2 teachers’ emotions. Fourth, the prevalence of females in our sample is a result of convenience sampling, which might limit the generalizability of our findings. To mitigate gender bias, subsequent research should aim to enlist a more balanced number of male and female L2 teachers. Fifth, we did not account for potential correlations among the four dimensions of SWE. This oversight might limit the credibility of our results. Future research should explore them as interrelated components. 

## 6. Conclusions

This study examined the effect of a SWE on L2 teachers’ emotions within the Chinese educational setting. The findings revealed that PC, PGI, and POS significantly enhanced enjoyment. However, SR did not noticeably affect this emotion. Furthermore, PGI and POS were found to significantly reduce anxiety, whereas PC and SR showed no such influence. In terms of pride, PGI alone had a significant positive effect. Lastly, POS was the only dimension that significantly decreased anger, with the other three dimensions showing no notable impact.

## Figures and Tables

**Table 1 behavsci-14-00370-t001:** Demographic information of the participants (*n* = 406).

Variables	Category	Frequency	Percentage
Gender	Male	62	15.27%
	Female	344	87.43%
Age	20–30	106	26.10%
	31–40	128	31.53%
	41–50	123	30.30%
	>50	49	12.07%
Educational level	Bachelor	163	40.15%
	Master	200	49.26%
	Ph.D.	43	10.59%

**Table 2 behavsci-14-00370-t002:** Confirmatory factor analysis.

Variables	Dimensions	Items	Outer Loadings	AVE	CR
SWE	PC	PC 1	0.860	0.760	0.905
		PC 2	0.886		
		PC 3	0.869		
	SR	SR 1	0.754	0.693	0.918
		SR 2	0.815		
		SR 4	0.830		
		SR 5	0.886		
		SR 6	0.871		
	PGI	PGI 1	0.778	0.603	0.901
		PGI 2	0.840		
		PGI 3	0.777		
		PGI 5	0.739		
		PGI 6	0.845		
		PGI 8	0.668		
	POS	POS 2	0.839	0.640	0.914
		POS 3	0.810		
		POS 4	0.805		
		POS 5	0.794		
		POS 7	0.790		
		POS 8	0.758		
Emotions	Enjoyment	Enjoy 1	0.865	0.695	0.901
		Enjoy 2	0.871		
		Enjoy 3	0.833		
		Enjoy 4	0.762		
	Anxiety	Anxiety 1	0.583	0.602	0.835
		Anxiety 2	0.837		
		Anxiety 3	0.875		
	Pride	Pride 1	0.694	0.589	0.849
		Pride 2	0.887		
		Pride 3	0.613		
		Pride 4	0.844		
	Anger	Anger 1	0.846	0.813	0.929
		Anger 2	0.923		
		Anger 3	0.934		

Note: SWE = supportive work environment; CR = composite reliability; AVE = average variance extracted; PC = perceived climate; SR = supervisory relationship; PGI = peer group interaction; POS = perceived organizational support.

**Table 3 behavsci-14-00370-t003:** Discriminant validity heterotrait–monotrait ratio.

Variables	1	2	3	4	5	6	7	8
PC	-							
2. SR	0.711	-						
3. PGI	0.663	0.718	-					
4. POS	0.800	0.844	0.626	-				
5. Enjoyment	0.604	0.556	0.571	0.560	-			
6. Anxiety	0.320	0.235	0.325	0.366	0.441	-		
7. Pride	0.571	0.502	0.482	0.476	0.776	0.341	-	
8. Anger	0.499	0.421	0.465	0.557	0.460	0.699	0.582	-

Note: PC = perceived climate; SR = supervisory relationship; PGI = peer group interaction; POS = perceived organizational support.

**Table 4 behavsci-14-00370-t004:** Descriptive statistics and correlation analysis.

Variables	Mean	*SD*	1	2	3	4	5	6	7	8
PC	3.390	0.850	-							
2. SR	3.468	0.789	0.778 *	-						
3. PGI	3.888	0.613	0.568 *	0.613 *	-					
4. POS	3.171	0.819	0.782 *	0.752 *	0.549 *	-				
5. Enjoy	3.072	0.507	0.517 *	0.474 *	0.474 *	0.493 *	-			
6. Anxiety	2.505	0.552	−0.306 *	−0.262 *	−0.305 *	−0.306 *	−0.412 *	-		
7. Pride	2.958	0.495	0.414 *	0.412 *	0.391 *	0.409 *	0.729 *	−0.328 *	-	
8. Anger	2.651	0.764	−0.270 *	−0.218 *	−0.225 *	−0.300 *	−0.331 *	0.860 *	−0.237 *	-

Note: PC = perceived climate; SR = supervisory relationship; PGI = peer group interaction; POS = perceived organizational support; Enjoy = enjoyment; * *p* < 0.05.

**Table 5 behavsci-14-00370-t005:** Regression for the effects of four dimensions of SWEs on the four emotions.

Hypotheses	Path	Std. Beta	SE	*p*-Value	Results	R^2^
H1	**PC ---> Enjoyment**	**0.233**	**0.080**	**0.004**	**SP**	0.329
	SR ---> Enjoyment	0.008	0.078	0.917	NS	
	**PGI ---> Enjoyment**	**0.247**	**0.057**	**0.000**	**SP**	
	**POS ---> Enjoyment**	**0.173**	**0.084**	**0.039**	**SP**	
H2	PC ---> Anxiety	−0.082	0.095	0.392	NS	0.156
	SR ---> Anxiety	0.080	0.106	0.450	NS	
	**PGI---> Anxiety**	**−0.211**	**0.065**	**0.001**	**SN**	
	**POS ---> Anxiety**	**−0.231**	**0.087**	**0.008**	**SN**	
H3	PC---> Pride	0.013	0.081	0.885	NS	0.074
	SR ---> Pride	0.010	0.083	0.907	NS	
	**PGI ---> Pride**	**0.163**	**0.058**	**0.005**	**SP**	
	POS ---> Pride	0.064	0.086	0.457	NS	
H4	PC---> Anger	−0.114	0.092	0.213	NS	0.108
	SR ---> Anger	0.102	0.097	0.296	NS	
	PGI ---> Anger	−0.106	0.063	0.094	NS	
	**POS ---> Anger**	**−0.239**	**0.079**	**0.003**	**SN**	

Note: PC = perceived climate; SR = supervisory relationship; PGI = peer group interaction; POS = perceived organizational support; SE = standard error; SP = significant and positive; NS = not significant; SN = significant and negative. Significant data were in bold.

## Data Availability

The data are available from the corresponding author upon reasonable request.

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
