# Peer review of "Exploring the Impact of a Supportive Work Environment on Chinese L2 Teachers’ Emotions: A Partial Least Squares-SEM Approach"

_behavsci, 2024, doi:10.3390/bs14050370_

Round 1
Reviewer 1 Report
Comments and Suggestions for Authors
The theme is interesting but the literature review is not exhaustive. Work in the field of emotions shows that positive emotions can also be harmful to learning. This is particularly what has been shown in the work calling on the model of Ellis and Moore, taken up by Ellis and Ashbrook according to which, pleasure leads to intrusive thoughts which require the allocation of attentional resources and which can impact the task in progress. The authors should be more controversial.
Generally speaking, the breakdown emotion by emotion in the introduction is strange. The authors should instead make a distinction between “effect of emotions on students” vs “effect of emotions on teachers”. For example, they could show the differentiated impact of emotions, whether positive or negative. This would avoid important oversights. For example, in the “Pride” subsection, the authors do not mention studies carried out directly with students.
After reading this first part, it also appears that the authors should mention the different dimensions of emotion: valence, intensity and dominance. This would certainly help to understand the rest. In addition, this would make it possible to understand the different impacts. Indeed, the studies cited sometimes evoke emotional valence, sometimes its intensity. It's not the same thing…
The same goes for the division relating to the work environment. Perhaps it would be better to articulate ideas rather than juxtapose them? Concerning SR, the subparagraph adds nothing. What studies have been carried out and what is the controversy about it?
Women are overrepresented in the sample. Is this a bias? If we are interested in the impact on emotion, it is very possible and it would be appropriate to explain it.
Data analysis is complete.
Author Response
Dear Reviewer 1,
We are quite grateful for your constructive comments. We have thoroughly considered each of your suggestions and made corresponding revisions to our manuscript. You comments helped improve the quality of manuscript tremendously. Thanks again for kindness. Please see our revisions in our uploaded "Review Comments and Author Response”, many thanks.

Reviewer 2 Report
Comments and Suggestions for Authors
This paper consists of an important contribution to the scholarship. The authors utilise PLS-SEM to model the effects of the supportive work environment on L2 teachers’ emotional reactions. The sample size is good enough for this technique, the description of the methods is thorough and gives the reader a clear understanding of what goes on, assumption testing has also been performed and the data analyses appear robust. Overall, I am happy to suggest publication provided that the authors consult my comments below and implement the following revisions.
1. In section 2.2.2., where the authors discuss the importance of the organisational support, they can cite the beneficial correlation between administrative support and coping as well as teachers’ self-efficacy and motivation:
Özdemir, G. (2020). The Mediating Role of Perceived Administrative Support for the Effect of Job Motivation on Organizational Identification. International Journal of Evaluation and Research in Education, 9(3), 539-547.
Katsantonis, I. (2020). Teachers’ self-efficacy, perceived administrative support and positive attitude toward students: Their effect on coping with job-related stress. Hellenic Journal of Psychology, 17(1), 14. https://doi.org/10.26262/hjp.v17i1.7843
2. Why are there no research hypotheses? The field is rich enough to warrant some hypotheses and this study is more confirmatory than exploratory in that sense.
3. Is the estimation based on the PLS-SEM or the consistent PLS-SEM available in SmartPLS? It should be the latter but please clarify.
4. Although I am okay with the adopted PLS-SEM approach, I would suggest the authors to give some more details of how their study’s design is more suitable for PLS-SEM rather than CB-SEM.
5. PLS-SEM has developed fit indices (e.g., SRMR, exact fit, NFI), but these indicators of good/bad fit of the models have not been presented in the paper. I would advise the authors to report these indices to gauge how good approximation is the model to the data.
6. Since the authors know that Cronbach’s alpha has several limitations (e.g., tau-equivalence), then why do they still report this measure instead of Composite Reliability?
7. In line 304 the authors mentioned that ‘all variables have positive correlations’; however, this is not true according to Table 4- some negative correlations are shown in Table 4.
8. Does the structural model account for the correlations between the exogenous variable (SWE components) and the disturbances of the outcomes?
9. A path diagram with the final results would be really helpful in section 4.3 to clarify the relations in a quick visual manner.
10. In the first paragraph of the discussion, I would recommend discussing the importance of this study before delving deeper into the methodological nuances.
11. Where the limitations of the study are brought up, I would suggest to mention that the results are not generalisable due to the convenience sampling design. Also, more longitudinal research is needed.
12. The conclusion section could be written as a flowing narrative conclusion rather than bullet points. Now, it feels like the authors were in a hurry.
In general, the paper is well-cited, but the iThenticate similarity report shows some overlap with previous works, especially concentrated in the definitions. I strongly recommend paraphrasing/ revising the following lines:
55-56, 60-65, 113-114, 119-122, 125-130, 218-226, 228, 229-230, 238-239.
I hope the authors will find the above comments helpful! I look forward to reviewing the revised version soon.
Author Response
Dear Reviewer 2,
We hope you are doing well today. We did appreciate your constructive comments, which improved the quality of our manuscript greatly. We have thoroughly considered each of your suggestions and made corresponding revisions to our manuscript. Please see our revisions in our uploaded "Review Comments and Author Response". Thanks again for your expertise and kindness.

Round 2
Reviewer 2 Report
Comments and Suggestions for Authors
I congratulate the authors on the hard work invested in providing a revised version of this manuscript. This article is a nice and robust contribution to the literature. I have now concluded my review of the revised version and I can say that they authors have provided convincing responses to my initial comments. The analyses are now much more robust and the presentation of the results and the literature review is much more appealing. The manuscript is now ready for publication. Just some final language issues that could be addressed during proofreading:
· Line 371: psychology expert in translation
· Line 397: predicts rather than forecasts
· Line 435: demonstrate rather than demonstrates
· Line 461: discriminant validity using the….
· Line 475: recorded at 0.92 rather than 0.092?
· Line 518: the effects of four dimensions…
· Line 503: the hypotheses were tested…
· Line 532: employees’ well-being…
Please also proofread the manuscript.
My best wishes for a swift publication!
Comments on the Quality of English Languagethe manuscript needs thorough proofreading after removing the track changes
Author Response
Dear Reviewer 2,
We sincerely appreciate the time and effort you have dedicated to reviewing our manuscript. Your insightful comments have been invaluable, and we are deeply thankful for your thorough and considerate feedback.
In response to your suggestions, we have carefully addressed the language issues you highlighted. The revisions have been clearly marked in blue in the updated version of our manuscript, which we have now uploaded for your convenience. Additionally, we have utilized Grammarly to ensure any potential grammatical errors have been rectified.
We hope this meets your approval, and we extend our best wishes for a pleasant day.
Warm regards
